# Spectroscopic identification of active sites for the oxygen evolution reaction on iron-cobalt oxides

Rodney D.L. Smith [1], Chiara Pasquini [2], Stefan Loos[2], Petko Chernev[2], Katharina Klingan[2], Paul Kubella[2], Mohammad Reza Mohammadi[2], Diego Gonzalez-Flores [2] & Holger Dau [2]

The emergence of disordered metal oxides as electrocatalysts for the oxygen evolution reaction and reports of amorphization of crystalline materials during electrocatalysis reveal a need for robust structural models for this class of materials. Here we apply a combination of low-temperature X-ray absorption spectroscopy and time-resolved in situ X-ray absorption spectroelectrochemistry to analyze the structure and electrochemical properties of a series of disordered iron-cobalt oxides. We identify a composition-dependent distribution of di-μ-oxo bridged cobalt–cobalt, di-μ-oxo bridged cobalt–iron and corner-sharing cobalt structural motifs in the composition series. Comparison of the structural model with (spectro)electrochemical data reveals relationships across the composition series that enable unprecedented assignment of voltammetric redox processes to specific structural motifs. We confirm that oxygen evolution occurs at two distinct reaction sites, di-μ-oxo bridged cobalt–cobalt and di-μ-oxo bridged iron–cobalt sites, and identify direct and indirect modes-of-action for iron ions in the mixed-metal compositions.

[1] Department of Chemistry, University of Waterloo, 200 University Avenue W., Waterloo, ON, Canada N2L 3G1. [2] FB Physik, Freie Universität Berlin, Arnimallee 14, 14195 Berlin, Germany. Correspondence and requests for materials should be addressed to R.D.L.S. (email: rodsmith@uwaterloo.ca) or to H.D. (email: holger.dau@fu-berlin.de)

Development of electrocatalysts for the oxygen evolution reaction (OER) comprised of earth-abundant materials is a key challenge in the global deployment of alternative, sustainable fuels[1–4]. Existing design principles for heterogeneous electrocatalysts can be largely attributed to the ability to tune the composition of crystalline materials in order to optimize a chemical property within a crystal structure family. This approach has revealed that descriptors, such as metal-oxygen bond strength[5] or the population level of antibonding orbitals[6] can successfully describe catalytic performance, and has ultimately led to the modern understanding of scaling relations in electrocatalysis[7–10]. The accuracy of design principles and predictive models, however, requires accurate structural models for the electrocatalyst material based on experimental identification of reaction sites and intermediates[11–15]. Reports of amorphization under electrocatalytic OER conditions introduces the possibility that the catalytically active species assumes a fundamentally different structure than the original material[16–18]. Development of robust structural models for disordered metal oxide phases are therefore expected to improve electrocatalyst design capabilities.

Disordered metal oxides have become a focal point in electrocatalyst research in recent years due to their tendency to outperform their crystalline analogs. Disordered cobalt oxide fabricated by anodic electrodeposition (CoCat) is the most thoroughly studied example in this class of materials[19–21]. X-ray absorption fine-structure spectroscopy (EXAFS) has been instrumental in developing structural models for CoCat, revealing a framework comprised exclusively of di-μ-oxo bridged Co–Co octahedra akin to layered double hydroxides (LDH) such as LiCoO2 or CoO(OH)[22,23]. An initially mixed $Co^{II}/Co^{III}$ state in electrodeposited films is oxidized to $Co^{III}$ in a precatalytic redox process[23,24], then to a mixed $Co^{III}/Co^{IV}$ state during OER catalysis[24,25]. Cobalt ions are redox-active throughout the film thickness[26] but changes in material fabrication conditions change the degree of order, affecting the electrochemical behavior of CoCat[27–29]. Characterization of similarly fabricated nickel[30,31] and manganese[32] oxides suggest the LDH model may be broadly

applicable, but it remains unresolved whether disordered oxides prepared by alternative techniques possess similar structures[33–35].

Herein, we address this issue via structural analysis of a series of photochemically deposited Fe–Co oxide films that were previously reported to exhibit composition-dependent trends in electrocatalytic OER performance[34,36]. We apply EXAFS to identify three distinct structural motifs that exist across the series in a composition-dependent fashion and utilize a selection of electrochemical and spectroelectrochemical experiments, including quasi in situ EXAFS and time-resolved in situ X-ray absorption spectroscopy (XAS), to establish trends in electrochemical behavior. Comparison of structural and behavioral trends leads us to propose the coexistence of multiple active sites for electrocatalytic OER. We propose a branching reaction mechanism that accounts for the observed behavior.

## Results

**Film fabrication.** A series of binary metal oxide films based on the formula $Fe_{100-y}Co_yO_x$ was prepared by photochemically induced decomposition of metal-organic precursor compounds[34]. Solid-state films consisting of mixtures of Fe(III) 2-ethylhexanoate and Co(II) 2-ethylhexanoate were prepared by spin-coating ethanolic solutions onto suitable substrates. Irradiation by UV-light yielded the desired metal oxide films. Films prepared by this approach have been characterized to exhibit thicknesses of 100–150 nm, possess Fe:Co atomic ratios identical to the precursor solutions and produce no reflections in X-ray diffraction studies[34,36,37]. Metal oxide films are labeled herein by the molar proportion of cobalt in the precursor solution (i.e., 100% Co, 88% Co, 75% Co, and so on).

**Electrochemical characterization.** Voltammetric experiments reveal a series of composition-dependent redox transitions that persist across the composition series. The initial anodic sweep on freshly prepared 100% Co films is characterized by oxidation processes with peak potentials of 0.97 ($E_{p,1}$), 1.17 ($E_{p,2}$), and 1.41 V vs. RHE ($E_{p,3}$), and an exponential increase in current

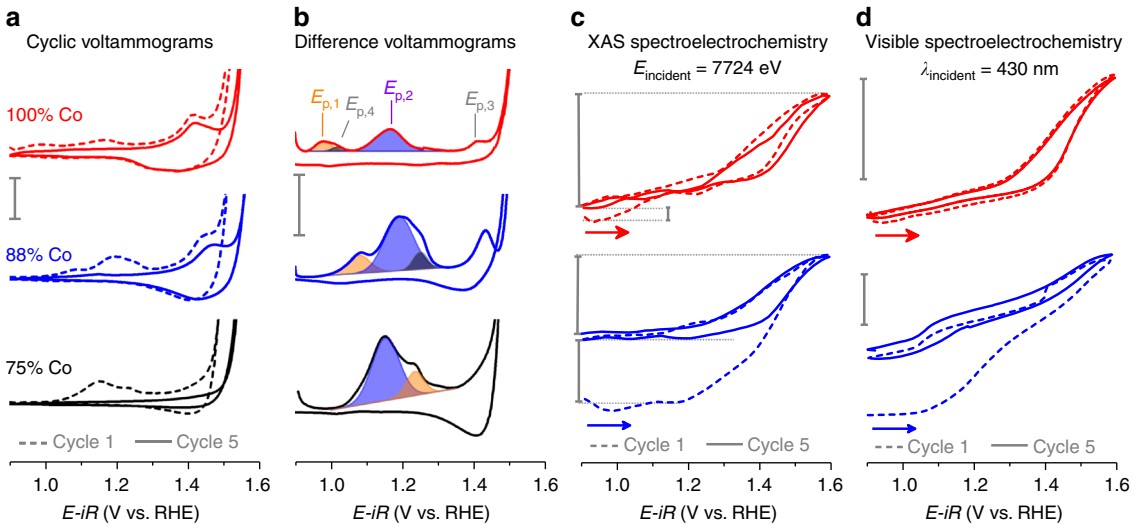

**Fig. 1** Electrochemical and spectroelectrochemical behavior of representative Fe–Co oxides. **a** The first and fifth voltammetric cycles recorded at 10 mV s$^{-1}$ in 1 M KOH$_{(aq)}$. Scale bar represents 1 mA cm$^{-2}$. **b** Difference voltammograms obtained by subtraction of cycle 5 from cycle 1; colored peaks depict the anodic shifting of the major irreversible components. Scale bar represents 0.2 mA cm$^{-2}$. **c** Shift in the Co- K-edge position ($\Delta E_{edge}$) during in situ X-ray absorption spectroscopy-cyclic voltammetry (XAS-CV) experiments for 100% Co and 88% Co. Scale bars, listed from top to bottom, represent 0.83 ($\Delta E_{edge,rev.}$), 0.05 ($\Delta E_{edge,irrev.}$), 0.67 ($\Delta E_{edge,rev.}$), and 0.45 eV($\Delta E_{edge,irrev.}$). **d** Change in transmittance during cyclic voltammetry experiments for 100% Co and 88% Co. Voltammetric data shown was acquired during visible spectroelectrochemical experiments. Scale bars represent 0.05 absorbance units. Data for the remaining compositions in series is available in Supplementary Fig. 1

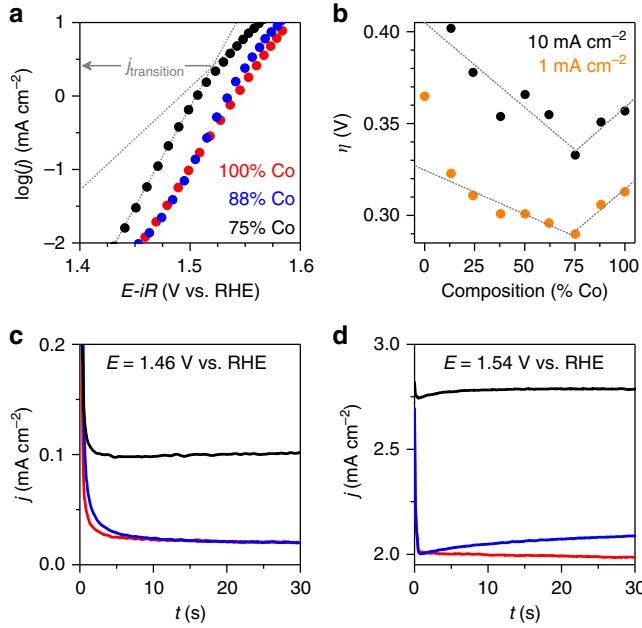

**Fig. 2** Electron transfer kinetics for representative Fe–Co oxides. **a** Tafel plots obtained from 60 s chronoamperometric measurements in 1 M KOH$_{(aq)}$. Trend lines are included for 75% Co to highlight the transition in Tafel slope. **b** Overall electrocatalytic OER performance as a function of film composition; trend lines shown to highlight the trends. **c**, **d** Current densities recorded following +10 mV potential steps to 1.46 and 1.54 V vs. RHE depicting the transient behavior in the two linear Tafel regimes from **a**

corresponding to electrocatalytic OER (Fig. 1a). The cathodic sweep exhibits a broad reduction process with a peak at 1.37 V and a shoulder at ~1.29 V. Subsequent voltage-cycling yields a loss of $E_{p,1}$ and $E_{p,2}$ and a decrease in the size of $E_{p,3}$. A difference voltammogram (Fig. 1b, Supplementary Fig. 1), obtained by subtraction of the current density of the fifth cycle, where the redox processes have stabilized, from the initial cycle, clearly resolves irreversible redox processes at $E_{p,1}$, $E_{p,2}$, and $E_{p,3}$ and a shoulder at 1.00 V ($E_{p,4}$) which exhibit composition-dependent behavior (Supplementary Table 1). With the exception of $E_{p,2}$, the irreversible processes shift anodically as Fe-content in the films is increased.

Analysis of electron transfer kinetics reveals trends between Fe-content and overall catalytic performance, as well as individual catalytic mechanism descriptors. A 44 mV dec$^{-1}$ Tafel slope was observed to extend to 10 mA cm$^{-2}$ for 100% Co (Fig. 2a). Two prominent changes occur upon addition of Fe: (i) the Tafel slope decreases as Fe-content increases, and (ii) a transition in Tafel slope occurs at ~1.53 V ($E_{transition}$), where the initial Tafel slope transitions to ~65 mV dec$^{-1}$. The decreased Tafel slope yields improved catalytic performance at low overpotentials. The current density at which the transition to a larger Tafel slope occurs ($j_{transition}$) decreases with Fe content, however, giving rise to a two-phase performance trend (Fig. 2b, Supplementary Fig. 2, Supplementary Table 2). Observation of a continuous Tafel slope for 100% Co rules out experimental concerns such as localized pH gradients as being responsible for the change in Tafel slope, suggesting a change in surface speciation, dominant reaction mechanism, or reaction site[38]. Reports of a Co$^{III}$/Co$^{IV}$ transition at 1.54 V vs. RHE for Co$_3$O$_4$[18], and at ~1.2 V vs. NHE for

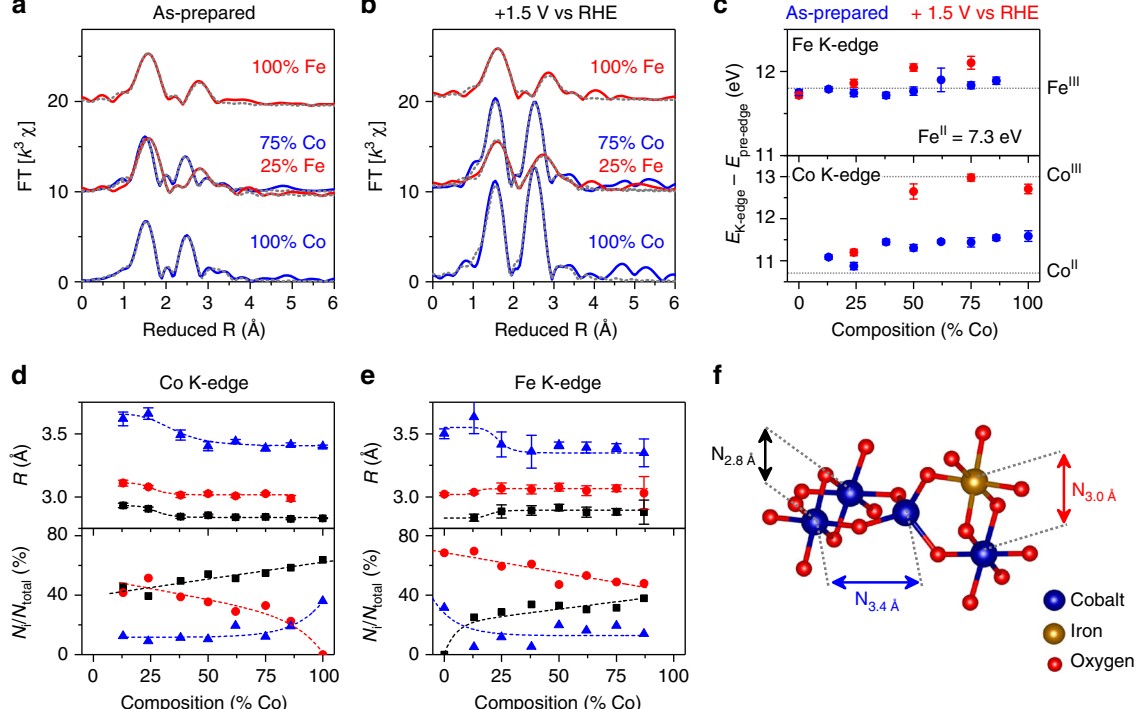

**Fig. 3** Structural analysis of the Fe–Co oxide by X-ray absorption spectroscopy. **a**, **b** Fourier-transform of the fine-structure region for Co and Fe K-edges in the as-prepared and oxidized states. Simulation results are represented by the dashed gray lines. **c** Difference in energy between the pre-edge feature and the K-edge for Fe and Co in the as prepared and oxidized states. Values obtained from crystalline reference compounds are marked (Supplementary Fig. 3). Error bars represent the standard deviation between all individual spectra for a given composition. **d**, **e** Bond lengths and relative structural contributions of the three unique M–M coordination shells for the Co and Fe K-edges of as-prepared catalyst films. Dashed lines are included to facilitate visualization of the trends. Error bars were calculated as described in Supplementary Note 3. **f** Hypothetical structure for visualization of the three distinctive Co–M structural motifs utilized in EXAFS simulations. Label colors in **f** correspond to data in **d**, **e**

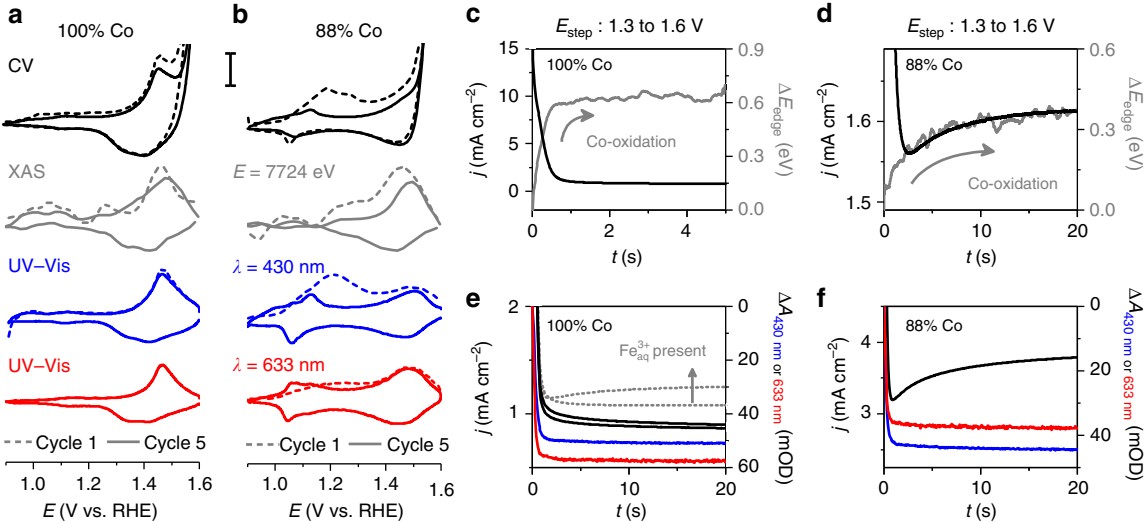

**Fig. 4** Spectroelectrochemical behavior of representative Fe–Co oxides. **a**, **b** Current density for cyclic voltammetric (CV) experiments and the derivative of spectroscopic signals for in situ X-ray absorption spectroscopy (XAS) and UV–visible spectroscopy (UV–vis) measurements. Scale bar represents 0.2 mA cm$^{-2}$. **c**, **d** Comparison of current density and change in Co K-edge location during in situ XAS measurements following anodic potential steps. **e**, **f** Comparison of current density and visible light transmittance recorded during anodic potential steps. Current densities and changes in absorbance are shown for the tenth anodic step on electrodes examined in electrolyte solutions that were purified of Fe$^{3+}$ impurities. The first and 10th anodic potential steps are shown for a 100% Co electrode that was examined in an electrolyte solution that was not purified of Fe$^{3+}$ contaminations **e**

electrodeposited cobalt oxide (pH 7)[24] leads us to assign $E_{transition}$ to the emergence of a Co$^{III/IV}$ oxidation process as a dominant factor in the catalytic mechanism. Inspection of chronoamperometric (CA) data underlying the Tafel plots reveals that the change in Tafel slope is accompanied by a change in transient current behavior. The current density decays exponentially with time at voltages below $E_{transition}$ (Fig. 2c), while an initial exponential decay in current density transitions to a slower exponential growth at voltages above $E_{transition}$ (Fig. 2d).

**X-ray absorption near-edge spectroscopy.** The K-edge locations measured by low-temperature XAS suggest that iron exists as Fe$^{III}$ while cobalt coexists as a composition-dependent blend Co$^{II}$ and Co$^{III}$ across the composition series. The Co K-edge in as-prepared films was observed to shift to lower energies as Fe-content in the catalyst films increased–indicative of a decrease in average oxidation state (Fig. 3c)[28]. Comparison of the −0.7 eV shift observed between 100% Co and 13% Co with the 2.3 eV per oxidation state shift observed for crystalline reference compounds (Supplementary Fig. 3a) suggests that Fe-incorporation stabilizes approximately one third of Co atoms in the catalyst films as Co$^{II}$ rather than Co$^{III}$. Direct comparison of K-edge locations to reference compounds suggests the average Co oxidation state decreases from ~Co$^{+2.4}$ in 100% Co to Co$^{+2.1}$ in 13% Co. Quasi in situ XANES spectra acquired on select catalyst films reveal a positive shift in the Co K-edge location at the catalytically operational voltage of 1.5 V vs. RHE, with 100% Co, 75% Co, and 50% Co approaching complete conversion to Co$^{III}$. In contrast, the Fe K-edge remained constant between 7124.7 and 7124.8 eV across the composition series (Fig. 3c, Supplementary Fig. 3). Comparison of the minor variations in edge position with crystalline reference compounds, which exhibit a 4.6 eV per oxidation state shift in K-edge location (Supplementary Fig. 3d), lead us to conclude that Fe$^{III}$ is predominant in all films.

**X-ray absorption fine-structure spectroscopy.** Simulations of the EXAFS spectra reveal that the local structure around both Co and Fe can be described by a limited number of structural motifs

that exist in a composition-dependent fashion across the composition series (Fig. 3, Supplementary Figs. 4, 5, Supplementary Tables 3, 4). The minimal-fit model for 100% Co consists of four coordination shells, each defined by a unique interatomic distance ($R_i$) and average coordination number ($N_i$). A Co–O shell ($R_{1.9 Å} = 1.91 Å$; $N_{1.9 Å} = 4.1$) and a Co–Co shell ($R_{2.8 Å} = 2.83 Å$; $N_{2.8 Å} = 2.1$) provide the fingerprint of di-μ-oxo bridged cobalt octahedra–the dominant structural motif in LDH structures and CoCat[22,23]. A Co–Co shell at 3.40 Å ($N_{3.4 Å} = 1.2$) represents ~1/3 of Co–Co vectors in 100% Co. A tetrahedral Co$^{II}$ bridging a di-μ-oxo bridged octahedra yields distances of 3.40 Å in Co$_3$O$_4$ and 3.45–3.49 Å in Fe$_2$CoO$_4$[18,39,40]. We therefore couple the 3.40 Co–Co shell with the second Co–O coordination shell ($R_{2.1 Å} = 2.07 Å$; $N_{2.1 Å} = 1.3$) and assign the pair of shells to corner-sharing Co$^{II}$ motifs ($N_{3.4 Å}$ in Fig. 3f). Quasi in situ measurements reveal that the Co–Co distance for di-μ-oxo bridged Co octahedra remains constant ($R_{2.8 Å} = 2.84 Å$) upon oxidation, while the Co–Co distance in corner-sharing motifs contracts ($R_{3.4 Å} = 3.37 Å$). The two Co–O shells merge into a single Co–O coordination shell (1.89 Å, $N_{1.9 Å} = 5.9$), suggesting that corner-sharing Co$^{II}$ motifs are oxidized to Co$^{III}$.

Simulations of Co K-edge EXAFS spectra for Fe-containing samples require addition of a Co–Fe shell to the 100% Co model. The shell exhibits a constant interatomic distance ($R_{3.0 Å} = 3.0 Å$) between 88% Co and 38% Co, matching that of di-μ-oxo bridged Co$^{II}$–Fe$^{III}$ octahedra in the spinel Fe$_2$CoO$_4$[39,40]. Iron-incorporation simultaneously increases the coordination number for the Co–Fe coordination shell ($N_{3.0 Å}$) and decreases the di-μ-oxo bridged Co–Co shell ($N_{2.8 Å}$), revealing that Fe$^{III}$ directly substitutes Co ions residing in the di-μ-oxo bridged Co–Co structural motif. The decrease in average Co oxidation state upon addition of Fe (Fig. 3c) indicates a di-μ-oxo bridged Fe$^{III}$–Co$^{II}$ motif. All coordination shells experience an abrupt change in interatomic distances at 25% Co, becoming comparable to those observed in photochemically deposited iron oxide. We surmise that Co atoms become interspersed in an iron-oxide matrix in these compositions. Similar trends are observed in Fe coordination environments as discussed in Supplementary Note 3. The prevalence of each of the three distinct structural motifs as a

function of Fe-content is provided in Fig. 3d as the Co–M coordination number relative to the total Co–M vectors (i.e., $N_i/N_{total}$).

**Spectroelectrochemical analysis**. In situ spectroelectrochemical-CV experiments were performed using both X-ray radiation and visible light. The derivative of spectroscopic signals recorded during these spectroelectrochemical-CV experiments contain peaks co-located with those observed in the voltammetric data. The reversible oxidation and reduction peaks are resolved by both types of measurements, while irreversible redox processes are resolved only under 430 nm irradiation (Fig. 4a, b). While in situ XAS-CV experiments failed to clearly identify the irreversible redox processes, their assignment as Co-based transitions is nonetheless supported by irreversible changes in Co K-edge location during in situ XAS-CA experiments (Supplementary Figs. 6–14) and in situ XAS-CV experiments (Fig. 1c, Supplementary Fig. 15). Spectroelectrochemical measurements enable deconvolution of redox changes from catalytic OER currents[24], enabling the cobalt oxidation process to be tracked after it becomes impossible in electrochemical data. This approach indicates anodic shift rates of 1–3 mV per %-Fe for the reversible co-oxidation process, matching the electrochemically estimated shift for $E_{p,3}$ (Fig. 4).

In situ XAS-CA experiments were performed on catalyst films by stepping the voltage between 1.30 and 1.60 V vs. RHE. The exponential decay in current density recorded during anodic and cathodic CA experiments on 100% Co are tracked by changes in Co K-edge location, with both equilibrating within 2 s (Supplementary Fig. 10). Anodic voltage steps yield an exponential increase in both Co K-edge location and an increase in visible light absorbance with time (Fig. 4). As K-edge location is directly correlated to average cobalt oxidation state, the positive shift in K-edge signifies a net oxidation of cobalt ions in the film. An exponential decrease in K-edge location during cathodic voltage steps signifies a net reduction of cobalt atoms (Supplementary Fig. 10). Addition of Fe to catalyst films resulted in emergence of the biphasic behavior that was observed during Tafel analysis, where a rapid exponential decay in current density gives way to a slower exponential growth during anodic voltage steps (Fig. 4d). The Co K-edge location shifts exponentially with time for these compositions, but requires ~30 s to equilibrate (Supplementary Figs. 11–14). The shift in K-edge position thus mirrors both phases observed in the CA results (Fig. 4d; Supplementary Fig. 11), confirming that cobalt-based oxidation processes underlie both transient phases. The crucial role of Fe in the biphasic behavior is confirmed by emergence of the biphasic behavior in 100% Co films when examined in unpurified electrolyte solutions, but not in solutions purified of Fe-contaminants (Fig. 4e). Although visible light spectroelectrochemistry resolved the irreversible redox processes during CV experiments, it revealed no spectral changes associated with the biphasic behavior in CA experiments (Fig. 4e, f).

**Structure-property trends**. The structural model generated for 100% Co is comprised of structural motifs reminiscent of the spinel crystal structure observed in $Co_3O_4$. This differs from the disordered layered-double hydroxide structure reported for electrodeposited cobalt oxides, revealing an influence of fabrication protocols on the localized structure even for highly disordered materials. Comparison of the electrochemical behavior with Co-based oxides in the literature reveals a clear influence of the short-range structure. A single-reversible redox transition (1.41 V) and four irreversible oxidation processes are observed for 100% Co (0.97, 1.00, 1.17, and 1.41 V vs. RHE) while $Co_3O_4$

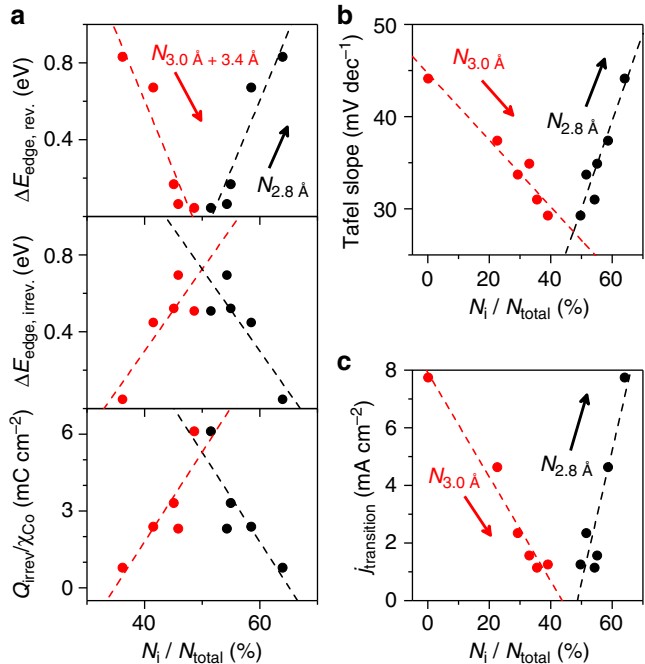

**Fig. 5** Correlations between structural features and electrochemical behavior parameters. **a** The change in the Co K-edge position in the reversible and irreversible regions of in situ XAS-CV experiments and the normalized charge density for irreversible redox processes ($Q_{irrev}/\chi_{Co}$). **b** Tafel slope. **c** Current density where transition between Tafel slopes occurs (Fig. 2)

exhibits a single-reversible redox transition (1.54 V vs. RHE)[18], anodically electrodeposited $Co(OH)_X$ exhibits two reversible redox transitions (~1.46 and 1.63 V vs. RHE)[24]; and cathodically electrodeposited $Co(OH)_2$ exhibits a single reversible (~1.10 V vs. RHE) and a single-irreversible transition (~1.05 V vs. RHE)[41]. The Tafel slope measured for 100% Co (44 mV dec$^{-1}$) is also lower than reported for $Co(OH)_2$ (~60 mV dec$^{-1}$)[41], $Co_3O_4$ (65 mV dec$^{-1}$)[18] or $Co(OH)_X$ (~60 mV dec$^{-1}$)[19].

Comparison of the proportion of structural motifs in the as-prepared films with (spectro)electrochemical behavior parameters across the composition series reveals a number of correlations that provide insights into the role of specific structural motifs (Fig. 5). Reversible oxidation state changes, estimated from the shift in Co K-edge position during in situ XAS experiments ($\Delta E_{edge, rev}$; Supplementary Fig. 15), yield a positive correlation with the proportion of di-µ-oxo bridged Co–Co motifs ($N_{2.8 Å}/N_{total}$; Fig. 5a). Quasi in situ XANES analysis indicates that an initially mixed $Co^{II}/Co^{III}$ state in 100% Co is converted near-quantitatively to $Co^{III}$ at catalytic potentials (Fig. 3c). We therefore assign the reversible redox process at 1.41 V to oxidation of $Co^{II}$ ions residing in di-µ-oxo bridged $Co^{II}$–$Co^{III}$ motifs. Irreversible Co oxidation-state changes ($\Delta E_{edge, irrev}$) and irreversible charge densities ($Q_{irrev}/\chi_{Co}$) yield positive correlations to the sum of corner-sharing Co and di-µ-oxo bridged Fe–Co motifs ($N_{3.0 Å + 3.4 Å}/N_{total}$; Fig. 5a). We tentatively assign $E_{p,2}$, the major irreversible process (Fig. 1b), to oxidation of $Co^{II}$ ions in di-µ-oxo bridged $Fe^{III}$–$Co^{II}$ motifs based on the importance of this feature in structural models (Fig. 3d; $N_{3.0 Å} > N_{3.4 Å}$). The second major irreversible feature ($E_{p,1}$) is thus assigned to oxidation of $Co^{II}$ ions residing in corner-sharing $Co^{II}$ structural motifs.

## Discussion

We propose the coexistence of two distinct electrocatalytic sites and suggest that Fe acts both directly and indirectly to catalyze OER in this composition series (Fig. 6). Tafel slopes have been

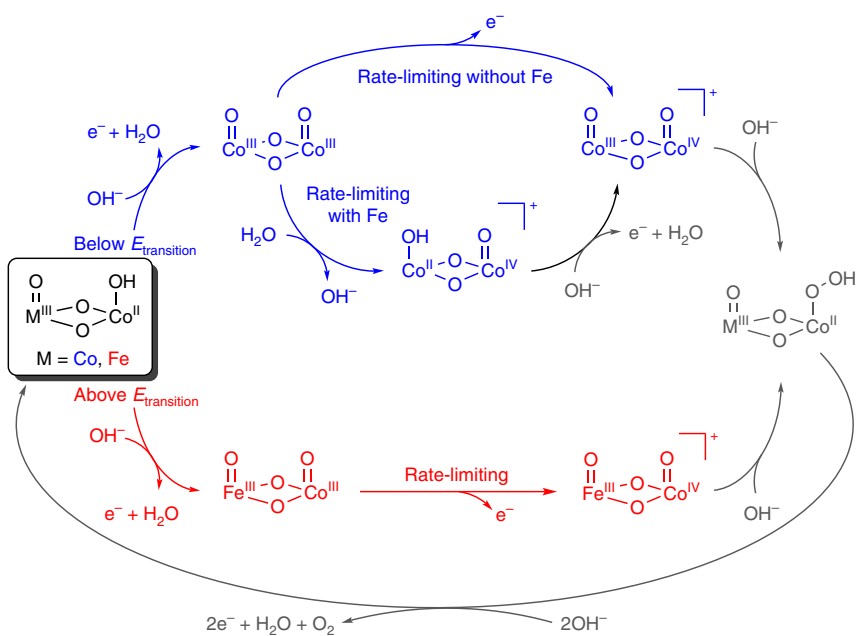

**Fig. 6** Electrocatalytic OER reaction mechanisms by photochemically deposited $Fe_{100-y}Co_yO_x$. Three distinct reaction sites are proposed to contribute to electrocatalytic OER across this composition series in a composition- and condition-dependent fashion

experimentally and theoretically shown to reflect reaction mechanisms and the rate-determining step in electrocatalytic reactions[38,42]. The 44 mV dec$^{-1}$ slope measured for 100% Co matches that predicted for a single-site reaction mechanism with oxidation of Co$^{III}$–Co$^{IV}$ as the rate-determining step[42]. This particular assignment yields a constant Tafel slope upon accumulation of Co$^{III}$, successfully describing the failure to observe a change in slope for 100% Co even at voltages where quantitative buildup of Co$^{III}$ is experimentally observed (Fig. 3c). Iron-incorporation induces a change in reaction profile, reflected by a correlation between Tafel slope and the proportion of di-μ-oxo bridged Co–Co motifs in the catalyst (Fig. 5b, Supplementary Fig. 2). A positive correlation between $j_{transition}$ and the proportion of di-μ-oxo bridged Co–Co motifs (Fig. 5c, Supplementary Fig. 2); however, reveals that di-μ-oxo bridged Co–Co motifs remain the dominant contributors to electrocatalytic OER below $E_{transition}$ and indicate an indirect role for Fe-ions. Insights into the nature of this role are gleaned from the Fe-induced anodic shift in $E_{p,3}$ (Fig. 1a), a Co$^{II/III}$ transition, and the lack of influence over $E_{transition}$ (Supplementary Fig. 2, Supplementary Table 2), a Co$^{III/IV}$ transition. Nocera and coworkers have reported an indirect role for Fe in electrocatalytic OER on Fe–Ni oxides wherein the high Lewis acidity of Fe$^{III}$ lowers the reduction potential for the Ni$^{III/IV}$ process[43], despite increasing the reduction potential of the Ni$^{II/III}$ process[44]. We propose that the unequal influence of Fe on the Co$^{II/III}$ and Co$^{III/IV}$ redox transitions observed here decreases the energetic barrier for disproportionation of di-μ-oxo bridged Co$^{III}$–Co$^{III}$ motifs in the oxidized catalyst film, thereby activating a secondary reaction pathway (Fig. 6). Equilibration of the Tafel slope at ~30 mV dec$^{-1}$ in the 25% Co–62% Co range of the composition series (Supplementary Table 2) provides excellent agreement with the predicted value for disproportionation as a rate-limiting step[42]. The observed accumulation of Co$^{III}$ in 75% Co by quasi in situ XANES near $E_{transition}$ (Fig. 3c) is predicted to impose a current density limit for di-μ-oxo bridged Co-Co sites[42], and emergence of biphasic transient behavior in CA data only when Fe is present and at voltages above $E_{transition}$ (Figs. 2, 4) reveals catalytic activity at di-μ-oxo bridged Fe–Co sites above $E_{transition}$. The biphasic

behavior observed in the presence of Fe can thus be interpreted as a convolution of the two distinct reaction sites, with $E_{transition}$ marking a transition between dominant reaction pathways. The sequence of reaction steps following the rate-limiting steps are drawn in Fig. 6 to be consistent with the relevant reaction mechanisms[42].

In summary, we have uncovered evidence for multiple reaction sites in a photochemically deposited $Fe_{100-y}Co_yO_x$ oxide series and analyzed their mechanistic roles and contributions towards electrocatalytic OER. Low-temperature XAS revealed that the composition series was comprised of a composition-dependent blend of di-μ-oxo bridged Co–Co, di-μ-oxo bridged Fe–Co and corner-sharing Co motifs. The presence of Fe as a component of the film or as an impurity in the electrolyte solution yielded an anodic shift in precatalytic redox features, decreased Tafel slopes at early stages of catalysis, induced a transition to a larger Tafel slope at a composition-independent voltage ($E_{transition}$) and triggered the emergence of biphasic chronoamperometric behavior above $E_{transition}$. Comparison of time-resolved in situ XAS spectroelectrochemistry results with the structural model enabled assignment of major voltammetric redox features to cobalt ions residing in unique coordination environments. Correlation between the proportion of di-μ-oxo bridged Co–Co sites and the current density at which $E_{transition}$ occurs indicates that electrocatalytic OER predominantly occurs at these sites below $E_{transition}$, even in the presence of Fe, while the biphasic behavior above $E_{transition}$ suggests convolution of multiple reaction sites. We propose a branching catalytic mechanism wherein the Fe-induced anodic shift in the precatalytic Co$^{II/III}$ redox transition activates disproportionation of di-μ-oxo bridged Co$^{III}$–Co$^{III}$ intermediates, introducing a reaction pathway with altered electrokinetic behavior. A secondary reaction site is then activated by oxidation of di-μ-oxo bridged Fe–Co sites at voltages above $E_{transition}$, yielding the observed biphasic behavior. These results provide a molecular-level understanding of this disordered heterogeneous electrocatalyst system and suggest that establishing synthetic control over the presence and population of specific coordination geometries may be a fruitful strategy in electrocatalyst design.

## Methods

**Film preparation**. Fluorine-doped tin oxide coated glass (FTO; TEC-7 grade, Solems S.A.) and glassy carbon wafers (GC; Sigradur K grade, Hochtemperatur-Werkstoffe GmbH) were cleaned by sequential ultrasonication in deionized $H_2O$ then ethanol for 15 min each. Substrates were dried under a stream of $N_2$ and placed in a UV-chamber for 15 min. The composition of photochemically deposited metal oxide films has been shown to match that of the precursor solution utilized[34,36]. The $Fe_{100-y}Co_yO_x$ series (y-values from 0 to 100 in ~12% steps) was thus prepared by dissolution of the appropriate ratio of cobalt (II) 2-ethylhexanoate (65% solution in mineral spirits, Strem Chemicals Inc.) and iron (III) 2-ethylhexanoate (6% solution in mineral spirits, Strem Chemicals Inc.) in ethanol to yield a total metal ion concentration of 0.3 M. Aliquots (10 μL) of precursor solutions were cast onto freshly cleaned substrates while spinning at 3000 r.p.m. (FTO substrates used for UV–visible spectroscopy, GC substrates used for XAS and electrochemistry). Subsequent irradiation by UV light (M1 UV-Chamber, Dinies Technologies GmbH) for 16 h yielded the metal oxide films.

**Electrochemical measurements**. Electrochemical measurements were carried out using a Biologic SP300 potentionstat. Experiments were performed in Teflon (in situ X-ray absorption spectroscopy) cells designed in-house. Aqueous 1 M KOH electrolyte solutions were prepared with milli-Q $H_2O$ (18.2 MΩ) and reagent grade KOH. Iron contamination was removed from the solution by aging on a suspension of $Co(OH)_2$ and subsequent removal by centrifugation[41,45]. A Hydroflex reversible hydrogen electrode (RHE; Gaskatel GmbH) was used as a reference electrode and a Pt mesh as the counter electrode. Electrochemical data were acquired by sequentially performing (i) a series of five cyclic voltammetric sweeps, (ii) the desired chronoamperometric measurements, and (iii) a repetition of the CV cycling. Stability and reproducibility in electrochemical behavior of the catalyst films was ensured by comparing CV traces from steps (i) and (iii). Cyclic voltammetry experiments were performed by sweeping the voltage between the initial open-circuit potential (0.89–0.90 V for the series) and +1.6 V vs. RHE at 10 mV s$^{-1}$. Step (ii) consisted of a series of 60 s chronoamperometric experiments in 10 mV steps between 1.0 and 1.6 V for generation of Tafel plots, or alternating 2 min voltage steps between 1.3 and 1.6 V for spectroelectrochemical experiments. Solution resistance values of < 5 Ω were measured by electrochemical impedance spectroscopy and not compensated for during experiments.

**Spectroscopic measurements**. X-ray absorption spectroscopy experiments were performed on the KMC-3 beamline at the BESSY II synchrotron facility in Berlin, Germany. Detailed descriptions of data acquisition and analysis are provided in the Supplementary Notes 2, 3. UV–visible spectroelectrochemical measurements were performed in transmission mode using an in-house set-up as described in Supplementary Note 4.

**Data availability**. The data that support the findings of this study are available from the corresponding authors upon reasonable request.

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

## Acknowledgements

Financial support was provided by the Bundesministerium für Bildung und Forschung (BMBF) through grants MEOKATS, CO2EKAT and IN SITU-XAS), and by the Deutsche Forschungsgemeinschaft through the Berlin cluster of excellence, UniCat. We thank the Helmholtz-Zentrum Berlin for allocation of time on the KMC-3 synchrotron beamline (BESSY II synchrotron, Berlin, Germany). R.D.L.S. acknowledges financial support from the Alexander von Humboldt Foundation.

## Author contributions

R.D.L.S. and H.D. designed the study, analyzed the data and wrote the manuscript. C.P., S.L., and P.C. assisted in data analysis. C.P., S.L., P.C., K.K., P.K., M.R.M., and D.G.-F contributed to X-ray absorption spectroscopy data acquisition.

## Additional information

**Competing interests:** The authors declare no competing financial interests.

