## [Peer Review File · Nature Communications]

Reviewers' comments:

Reviewer #1 (Remarks to the Author):

In this manuscript the authors claim a compositional dependent formation of di-oxo bridged Co-Co and Co-Fe with corner-sharing Co structural motifs in the composition series as active sites for oxygen reduction. These claims are made on the basis of both ex situ and in situ XAS measurements.

However, there are fatal issues with the XAS data analysis as elaborated below, which makes the story unreliable.

1. It is impossible to fit the Fe-Fe or Co-Co with the bond distance of 3.4-3.5 Å because the corresponding signals are too weak (Figure 3b and 3e) (relative amplitude is around 28%). Actually, as can be seen in Table S3 and S4, the fitted values of the coordination number of Co-Co or especially Fe-Fe with the bond distance of 3.4-3.5 Å is comparable to the uncertainties. The authors claimed that including this path significantly improve the fitting quality, but that is likely caused by the increase of the variables.

2. The Ft-EXAFS data were fitted by including 5 paths, and correspondingly at least 12 variables have to be used for fitting. However, the number of independent points according to the range of k-space and R-space are around 19 for the Co edge and 13 for the Fe edge. Therefore, my concern would be that the number of variables are too close to the number of independent points (thumb rule is half), which basically can fit any data successfully.

3. The range of $0.5 \leq R (\text{Å}) \leq 4.0$ was used to fit the Co edge data. The lower end is too low way into the background region. EXAFS does not allow fitting with the lower end of R-space range smaller than $rbkg$, which is usually the half of the shortest bond distance and normally greater than 1. 0.5 is too small.

4. The Co-Fe bond distance is not equal to the Fe-Co bond distance (Table S3 and S4), which is wrong because they are essentially the same bond in XAS. The authors should fit the Fe and Co K-edge data simultaneously by using the same variable for both paths. This also applies to the debye-waller factor too.

5. I strongly question the validity of the way that authors separate the Fe-O paths with the bond distance around 2 Å when carrying out EXAFS fitting because the bond distances are too close to be distinguished by XAS. Although this way give better fitting parameters (R-factor), it is mostly caused by the increase of the variables and the results are not meaningful. A normal way is to fit it with only one path the slightly different bond distance will be reflected by the debye-waller factor. Similarly, Fe-Fe and Fe-Co (or Co-Co and Co-Fe) with similar bond distance cannot be distinguished by XAS as the author did since XAS cannot distinguish Fe from Co as surrounding atoms.

Overall, according to my experience, the XAS data presented in this work are very close to those of the Co or Fe oxide references, which can be easily verified by comparing the XANES and Ft-EXAFS spectra. Thus, any information beyond that has to be confirmed by complementary techniques. Reliable EXAFS fitting should only include the M-O ($\sim 2\text{Å}$) and M-M ($\sim 3\text{Å}$) paths. Anything beyond that has reliability issues.

Reviewer #2 (Remarks to the Author):

The authors prepared a series of disordered Fe/Co oxides through photochemical deposition, and applied low-temperature X-ray absorption spectroscopy (XAS) and time-resolved in-situ XAS spectroelectrochemistry to conduct thorough study on various Co structural motifs in the oxides and their mechanistic roles in catalyzing oxygen evolution reaction (OER). Through their experiments, they identified the contributions of different motifs to the voltametric redox processes, confirmed the active sites for OER, illustrated the roles of Fe ions on OER catalysis, and proposed OER mechanisms for the as-prepared oxides. The work is very interesting and could provide a profound understanding of the mechanisms of disordered electrocatalysts towards OER

catalysis. However, a major revision may be needed before acceptance:

1. Authors may need to provide XRD profiles of a series of their as-prepared Co-Fe oxides to demonstrate that they are disordered oxides.
2. Line 19, Page 2. Are there any evidence to prove that the 5th cycle are the behaviour at steady state?
3. Did the authors conduct the voltammetry study on a rotating electrode? If not, the "steady - state" as aforementioned may be inappropriate to use, and it would be better to illustrate how the authors rule out the possible effects from mass transfer on various samples.
4. Figure 1b. I would like to know more details about why the $E_{p,1}$ in 75% Co shifts to a potential even higher than $E_{p,2}$. Could the authors elaborate more details on how to assign the peaks to the specific oxidation processes?
5. Figure 3e. I think it may be inconvincible to use a linear line to fit the N3.0A data, otherwise it yields a nearly 40% N3.0A/Ntotal in 100% Co sample, where there should be no Fe ions.
6. Scheme 1. I suggest the authors to discuss more about why Co(IV) is reduced to Co(II) before the final step where O₂ is formed.
7. Some minor issues:
 - It would be easier to understand if the authors could use " $Fe_{1-y}Co_yO_x$ " instead of $Fe_{100-y}Co_yO_x$
 - Line 1-2, page 4. Any references to support it?
 - Figure 2b. " $\eta 1mA\ cm^2$ " is incorrect in unit of current density, and is not suitable to use due to the presence of data at $10mA\ cm^{-2}$.
 - Line 5-6, Page 5. It would be more convincible if the authors could provide the error band for valence of Co.

Reviewers' comments:

Reviewer #1 (Remarks to the Author):

In this manuscript the authors claim a compositional dependent formation of di-oxo bridged Co-Co and Co-Fe with corner-sharing Co structural motifs in the composition series as active sites for oxygen reduction. These claims are made on the basis of both ex situ and in situ XAS measurements.

However, there are fatal issues with the XAS data analysis as elaborated below, which makes the story unreliable.

Many of the reviewer's concerns center around the validity of our selected model. We modeled the system in a bottom-up approach and carefully compared numerous models along the way, including those stated by the reviewer to be the only reliable possibility. The only satisfactory description of the data arises from the model that is the focus of the manuscript. In consideration of the reviewer's concerns we have added information to the *Supplementary Information* to more clearly describe the fitting process and support the validity of our XAFS analysis. We have also made several smaller changes, as detailed in response to the individual comments. The following major changes were made to the Supplementary Information:

- Page S8, line 17 – The paragraph was edited to provide a more thorough description of our simulation process. The model was generated in a bottom-up approach and we have incorporated an explicit description and rationale for each step of the process.
- Fig. S16 – a new figure was added to enable readers to compare the XAFS data for **100% Co** with those for LiCoO₂ and electrodeposited cobalt oxide. Note that LiCoO₂ is the crystalline reference material most commonly used for the 2-shell model that the reviewer had concluded is the only valid model for the data. Numerous publications have identified the structure of electrodeposited cobalt oxide to be similar (as cited in the manuscript). Three regions of the data are marked on the figure to highlight the obvious differences in the data for **100% Co** that prevent use of this model.
- Figs. S17 and S18 – two new figures were incorporated to support the newly added discussion. These figures contain a selection of simulated models that were considered in our analysis of **100% Co** (Fig. S17) and **75% Co** (Fig. S18). The figures allow readers to see the unique features in the XAFS data present in our samples, and to observe how the simple 2-shell system completely fails to capture these key features. They also enable readers to readily see the effect of individual shells by observing the changes at each step of model development.
- Tables S5 and S6 – two new tables were added. These tables contain the fit parameters for the simulations shown in Figs. S17 and S18. Note that many of the parameters in the simple models contain physically unrealistic values: for example, the Co-O coordination number for the 2-shell fit suggested by the reviewer (fit labeled 100Co[1Co,1O]) indicates a Co-O coordination number below 4. Allowing the Debye-Waller parameter to be fitted does little to help – the Co-O coordination number remains well below 5 while the Debye-Waller parameter increases to 0.071 Å – a value large enough for an oxygen shell to indicate an inaccurate model.

1. It is impossible to fit the Fe-Fe or Co-Co with the bond distance of 3.4-3.5 Å because the corresponding signals are too weak (Figure 3b and 3e) (relative amplitude is around 28%). Actually, as can be seen in Table S3 and S4, the fitted values of the coordination number of Co-Co or especially Fe-Fe with the bond distance of 3.4-3.5 Å is comparable to the uncertainties. The authors claimed that including this path significantly improve the fitting quality, but that is likely caused by the increase of the variables.

Beyond signal:noise concerns, the intensity of peaks in the FT representation of the data are dependent on numerous parameters and cannot be used to judge the validity of coordination shell existence. Their existence must be determined mathematically by fitting the data k -space. Figures S14 and S15 show the Fourier Transform of the data out to 10 Å to allow readers to see the noise level in the data. The peak in question is >3x the height of the noise – noise is therefore not an issue.

The newly incorporated data for a selection of structural models for **100% Co** and **75% Co** (Figs. S17 and S18, and Tables S5 and S6) illustrates that the k -space data cannot be properly fitted without the use of this additional feature. The fits are given in both graphical and numeric formats as visual comparison of the fits often grants insights that are not observable in generalized fit-quality parameters such as the R-factor – fitting one portion of the data exceptionally well and another portion poorly will yield a “good” mathematical measure of fit quality but will result in a clearly flawed model.

The newly added comparison between **100% Co** and two materials with a layered-double hydroxide structure (LiCoO₂ and an electrodeposited cobalt oxide; Fig. S16) makes it abundantly clear that **100% Co** exhibits a substantially different structure, and that a simple 2-shell model cannot describe this system. Addition of a second Co-O distance and the 3.4 Å Co-M feature are required to provide an adequate fits of features in the data.

2. The Ft-EXAFS data were fitted by including 5 paths, and correspondingly at least 12 variables have to used for fitting. However, the number of independent points according to the range of k -space and R -space are around 19 for the Co edge and 13 for the Fe edge. Therefore, my concern would be that the number of variables are too close to the number of independent points (thumb rule is half), which basically can fit any data successfully.

The “rule of thumb” was obeyed: the reviewer has miscounted the number of variables used for fitting.

The Co spectra that were relied upon for the mechanistic discussion were fitted in k -space between k -3 and k -12. The Nyquist Criterion approximates 20 independent data points for R -space analysis to 4 Å. The data were fitted with 4 shells (100% CoOx) or 5 shells (the remaining samples). The E_0 , σ and S_0^2 values were all fixed during the fitting protocol, in the manner described in the manuscript (on page S8, paragraph beginning on line 11 – note that this paragraph has been edited in the revised manuscript). There are therefore 8 and 10 variables being fitted, dependent on the sample.

Note also that by limiting R-space below 4 Å we are being conservative compared to other publications in the field (for examples, a $1 < R < 6$ range was used in 2010 J. Am. Chem. Soc. 132, 13692.)

3. The range of $0.5 \leq R \text{ (Å)} \leq 4.0$ was used to fit the Co edge data. The lower end is too low way into the background region. EXAFS does not allow fitting with the lower end of R-space range smaller than rbkg, which is usually the half of the shortest bond distance and normally greater than 1. 0.5 is too small.

The reviewer appears to have misread our descriptions. The data was fitted entirely in *k*-space, from *k*-3 to *k*-12 for Co K-edge spectra. The lower end of our fitting region (*k*-3) is well within accepted bounds. The R-space range of 0.5 to 4.0 Å was utilized for estimation of error in the fitted parameters only, performed *via* back-transforming as described in the referenced journal article (ref. #1 in Supp. Info). This range affects only the estimated error in the fitted parameters.

- We have updated Table S3 with error values calculated using an R-space range of $1 < R < 4 \text{ Å}$.

4. The Co-Fe bond distance is not equal to the Fe-Co bond distance (Table S3 and S4), which is wrong because they are essentially the same bond in XAS. The authors should fit the Fe and Co K-edge data simultaneously by using the same variable for both paths. This also applies to the debye-waller factor too.

The XAFS oscillations in the Fe K-edge data are only observed up to approximately *k*-9.5, restricting our ability to generate high-quality fit data for Fe (this is a previously observed issue for disordered FeOx – see 2011 J. Solid State Chem. 184, 1025). This fact, combined with Co ions being the electrochemically active portion of the films and the focus of this work, contributed to our decision to only rely on and utilize the Co K-edge data in our detailed analyses and discussions. We nonetheless included the Fe data for full transparency to readers. We contend that interpretations arising from the lower quality Fe K-edge data does not make the higher quality of the Co K-edge inaccurate or irrelevant. We decided against simultaneously fitting the Co and Fe data due to the vast differences in data quality. Rather than sacrifice quality in the Co data, we opted to fit each element independently to ensure maximum fit quality for Co.

- We have edited the discussion regarding the Fe K-edge data to include the reviewer's concerns (paragraph starting page S9, line 16).

5. I strongly question the validity of the way that authors separate the Fe-O paths with the bond distance around 2 Å when carrying out EXAFS fitting because the bond distances are too close to be distinguished by XAS. Although this way give better fitting parameters (R-factor), it is mostly caused by the increase of the variables and the results are not meaningful. A normal way is to fit it with only one path the slightly different bond distance will be reflected by the debye-waller factor. Similarly, Fe-Fe and Fe-Co (or Co-Co and Co-Fe) with similar bond distance cannot be distinguished by XAS as the author did since XAS cannot distinguish Fe from Co as surrounding atoms.

While EXAFS cannot rigorously resolve differences on the order of 0.1 Å, it is not uncommon to split a single shell with a high σ value into two peaks if the data necessitates it and there is chemical/physical rationale to support the assignment. A survey of iron oxide crystal phases reveals that two unique Fe-O bond lengths is an extremely common feature. Examples include:

Goethite: 1.93, 2.10 Å

Hematite: 1.91, 2.16 Å

Fe₂CoO₄: 1.95, 2.06 Å

Lepidocrocite: 1.98, 2.07 Å

Based upon the prevalence of two unique bond lengths in crystal structures, it is entirely reasonable to split the Fe-O shell as we have.

The Co K-edge data consistently shows physically unrealistic parameters when a single Co-O shell is utilized. This was alluded to on line 11, Pg. 8 in the Supplementary Information and the revised manuscript has additional data to support the statements (Tables S5 and S6). Splitting of the Co-O shell into two shells is thus justified in those cases as well.

In regards to the M-M distances: we remind the reviewer that the data was analyzed not for a single sample, but across a series of 9 related samples. There are clear features in the *k*-space data that are mismatched if the Co-Fe shell is left out, as seen in the newly added Figures S17 and S18. Observation of these features across the composition series (Figs. S14 and S15) indicates that they are not random fluctuations. Systematic errors due to instrumentation or data handling can be ruled out by comparison of the data on as-prepared films with oxidized films (e.g. Fig. S14a with S14c): the distinct features undergo a change upon oxidation, disappearing and adopting a structure more in line with LiCoO₂. This structural change is consistent with the analysis and discussion in our manuscript.

It is true that we cannot experimentally distinguish whether the atom residing in a coordination shell is an Fe or Co atom. We describe in the manuscript (paragraph beginning on page 6, line 13) that we assigned the identity of Co-Co and Co-Fe shells based on the coordination shell with the longer bond length being required only in the mixed-metal, binary compositions.

Overall, according to my experience, the XAS data presented in this work are very close to those of the Co or Fe oxide references, which can be easily verified by comparing the XANES and Ft-EXAFS spectra. Thus, any information beyond that has to be confirmed by complementary techniques. Reliable EXAFS fitting should only include the M-O (~ 2Å) and M-M (~ 3Å) paths. Anything beyond that has reliability issues.

The simple model that the reviewer suggests has been abundantly successful in ascribing a layered-double hydroxide structure to cobalt oxide films that are prepared by electrodeposition, but it fails to provide a mathematically acceptable fit for the films studied here even when all of the fitting parameters are left free. We will highlight once again that while the peaks in the Fourier Transform representation of the data can be used to guide model development, it is not scientifically valid to judge the existence of coordination shells present in a material by visual

inspection. The newly added Figure S16 provides the fit requested by the reviewer, while Figures S17-S18 and Tables S5-S6 illustrate how unreliable the two shell model is for the materials in this manuscript.

The materials were analyzed with a combination of *in-situ* XAS spectroelectrochemistry, *ex-situ* XAFS, electrochemical characterization, and *in-situ* UV-vis spectroelectrochemistry. The correlations observed between these different techniques provide the complementary data that the reviewer requests.

Reviewer #2 (Remarks to the Author):

The authors prepared a series of disordered Fe/Co oxides through photochemical deposition, and applied low-temperature X-ray absorption spectroscopy (XAS) and time-resolved *in-situ* XAS spectroelectrochemistry to conduct thorough study on various Co structural motifs in the oxides and their mechanistic roles in catalyzing oxygen evolution reaction (OER). Through their experiments, they identified the contributions of different motifs to the voltametric redox processes, confirmed the active sites for OER, illustrated the roles of Fe ions on OER catalysis, and proposed OER mechanisms for the as-prepared oxides. The work is very interesting and could provide a profound understanding of the mechanisms of disordered electrocatalysts towards OER catalysis. However, a major revision may be needed before acceptance:

1. Authors may need to provide XRD profiles of a series of their as-prepared Co-Fe oxides to demonstrate that they are disordered oxides.

The disordered nature of the materials is readily apparent in the weak intensity of the features in the X-ray absorption fine-structure data. Crystallinity in the materials would cause the oscillations in Figs. S14 and S15 to be more intense and better defined, as seen for crystalline LiCoO_2 in the newly added Figure S16. To clarify this for readers we have added the following line to the Supplementary Information:

- (Page 8, line 23) - "...comparisons between the XAFS oscillations for **100% Co**, LiCoO_2 and **CoCat** (Fig. S16). This comparison reveals differences in both oscillation intensity, indicative of a more disordered structure around cobalt ions in **100% Co**, and in interference patterns in three distinct regions."

2. Line 19, Page 2. Are there any evidence to prove that the 5th cycle are the behaviour at steady state?

The "steady state" terminology was used to convey the stability in the pre-catalytic redox processes. A figure is attached below that shows the first 5 voltammetric cycles for 100% Co, 88% Co and 75% Co as confirmation that these redox processes have stabilized.

3. Did the authors conduct the voltammetry study on a rotating electrode? If not, the “steady-state” as aforementioned may be inappropriate to use, and it would be better to illustrate how the authors rule out the possible effects from mass transfer on various samples.

The “steady state” terminology that we employed in regards to the cyclic voltammetry was intended to highlight that changes in the voltammetric redox behavior are no longer occurring. We did not intend to convey the term in the electrokinetic sense.

- We have edited the discussion (page 2, line 19) to clarify our meaning and eliminate potential confusion

4. Figure 1b. I would like to know more details about why the $E_{p,1}$ in 75% Co shifts to a potential even higher than $E_{p,2}$. Could the authors elaborate more details on how to assign the peaks to the specific oxidation processes?

A description of our assignment is available in the Supplementary Information in the paragraph starting on page 11, line 2.

5. Figure 3e. I think it may be inconvincible to use a linear line to fit the N3.0A data, otherwise it yields a nearly 40% N3.0A/Ntotal in 100% Co sample, where there should be no Fe ions.

The dashed lines are not mathematical fit lines, they are there to help readers more easily follow the trends in the data.

- to avoid such inferences the lines in Fig. 3 were edited so that they end at the final data point in each series

6. Scheme 1. I suggest the authors to discuss more about why Co(IV) is reduced to Co(II) before the final step where O₂ is formed.

The elementary step in question, formation of a peroxide bond, requires two oxygen atoms to be oxidized by 1 unit each. The Co(IV) converts to Co(II) because it is formally accepting these electrons from the oxygen atoms. That said, the specific sequence of events after a rate-determining step cannot be known with any confidence. On page 11 we use a Tafel slope analysis from reference 43 to assign rate-limiting steps; we opted to draw the reaction cycles to be consistent with the reaction relevant mechanisms in reference 43. To clarify this ambiguity and provide readers the option to learn more about the chemistry of the specific mechanisms, we have added the following line to the mechanism discussion (page 12, line 1):

- “The sequence of reaction steps following the rate-limiting steps are drawn in Scheme 1 to be consistent with the relevant reaction mechanisms.⁴³”

7. Some minor issues:

- It would be easier to understand if the authors could use “Fe-(100-y)%Co_yO_x” instead of Fe_{100-y}Co_yO_x
- Line 1-2, page 4. Any references to support it?

- Citations for references 14 and 24 have been added.

• Figure 2b. “ η 1mA cm²” is incorrect in unit of current density, and is not suitable to use due to the presence of data at 10mA cm⁻².

The 1 mA cm⁻² is a subscript that was meant to symbolize the overpotential required to achieve that current density.

- We have changed the y-axis label in Fig. 2b to simply “ η (V)”.

• Line 5-6, Page 5. It would be more convincing if the authors could provide the error band for valence of Co.

Other Changes:

- The sample labels for Fig. S15b were incorrect and have been updated.

REVIEWERS' COMMENTS:

Reviewer #1 (Remarks to the Author):

I have carefully read through the rebuttal letter from the authors. The authors have tried to respond to all the points raised earlier in our review and accordingly have made a lot of changes. I believe the article has been strengthened. While I stick to my points on some of the issues I brought out, it is pointless to further argue with the authors. At least the parts that are certainly incorrect were removed. In general, this is a sophisticated article in which XAS has been extensively used to determine the atomic structure. The proposed structural model is interesting. It can be published as it is.

Reviewer #2 (Remarks to the Author):

The paper has been revised accordingly, and can be accepted now.

REVIEWERS' COMMENTS:

Reviewer #1 (Remarks to the Author):

I have carefully read through the rebuttal letter from the authors. The authors have tried to respond to all the points raised earlier in our review and accordingly have made a lot of changes. I believe the article has been strengthened. While I stick to my points on some of the issues I brought out, it is pointless to further argue with the authors. At least the parts that are certainly incorrect were removed. In general, this is a sophisticated article in which XAS has been extensively used to determine the atomic structure. The proposed structural model is interesting. It can be published as it is.

Reviewer #2 (Remarks to the Author):

The paper has been revised accordingly, and can be accepted now.

Response:

Following the reviewer comments, no scientific changes were made to the manuscript. Stylistic changes are summarized in the cover letter and shown in detail in the final Microsoft Word document.